# Strength Use and Well-Being at Work among Teachers: The Mediating Role of Basic Need Satisfaction

**DOI:** 10.3390/bs14020095

**Published:** 2024-01-27

**Authors:** Cangpi Wei, Jiahe Su, Jingjing Zhao, Ke Ding, Feng Kong

**Affiliations:** 1School of Psychology, Shaanxi Normal University, Xi’an 710119, China; wcp373991@snnu.edu.cn (C.W.); 13718194517@163.com (J.S.); zhaojingjing1987@snnu.edu.cn (J.Z.); 2School of Psychology, Shenzhen University, Shenzhen 518060, China

**Keywords:** strength use, hedonic well-being at work, eudaimonic well-being at work, teachers, basic need satisfaction

## Abstract

Teachers’ well-being at work is an important indicator of their mental health. Strengths use has been identified as a significant predictor of enhanced well-being at work. However, there is a scarcity of studies that have examined the connection between teachers’ strengths use and well-being at work; thus, its underlying psychological mechanism is unclear. Therefore, this study aimed to explore the association between teachers’ strengths and well-being at work together with the mediating role of basic need satisfaction. A total of 374 university teachers completed a series of questionnaires on strengths use, basic need satisfaction, hedonic well-being, and eudaimonic well-being. The results showed that there were positive correlations between strength use and both types of well-being at work. Moreover, basic need satisfaction mediated the association between strength use and two types of well-being at work. The findings suggest that institutions should prioritize enhancing teachers’ ability to utilize their strengths and foster an environment conducive to such practices, thereby improving their workplace well-being.

## 1. Introduction

Teachers are the most important resource for the high-quality development of education. They are responsible for teaching students knowledge and for the prosperity of the country. Improving teachers’ well-being at work not only directly contributes to their mental health and work performance [1] but also promotes schools’ sustainable educational effectiveness and student growth [2]. Consequently, examining the factors contributing to teachers’ well-being is of great importance.

Since the emergence of positive psychology, the concept of strength use (SU) has gained prominence, denoting the extent to which individuals utilize their strengths in their daily lives [3]. SU has been found to be closely related to well-being in its general context [4], but there are few studies examining SU and the well-being at work of college teachers [5]. Therefore, in this study, we examined the association between SU and the teachers’ well-being at work and explored the underlying mechanisms from the perspective of self-determination theory.

### 1.1. SU and Well-Being at Work

Well-being is considered an optimal state of mind and involves living human life to the best. It is not just about feeling good; it is about working well and living a good life [6]. William [7] believed that well-being encompasses how people experience and evaluate their lives positively. In the literature [7], well-being was conceptualized through two foundational theoretical perspectives: the hedonic and eudaimonic. From the hedonic approach, well-being focuses on the pursuit of happiness and life satisfaction. The eudaimonic approach emphasizes the pursuit of the realization of human potentials [8]. Huang et al. studied the effect of job characteristics on hedonic well-being and concluded that trust in colleagues was positively correlated with teachers’ self-esteem and hedonic well-being [9]. In addition, Wang et al. analyzed the hedonic well-being and eudaimonic well-being of students through machine learning and traditional statistical analysis, and they obtained different influencing factors for the two types of well-being [10]. Thus, inspired by this method, we posit that teachers’ well-being at work refers to the sum of a teacher’s hedonic well-being at work and eudaimonic well-being at work. Several studies have separately explored hedonic and eudaimonic well-being at work. For example, Nina [11] demonstrated that hedonic well-being among female workers could be predicted by work–family balance and optimism. As for eudaimonic well-being at work, Kundi, et al. [12] found that eudaimonic well-being at work was related to job performance and affective commitment. Although many scholars have studied either kind of well-being at work, few studies have examined both kinds of well-being together in organizational psychology. Therefore, conducting research that compares and synthesizes the two kinds of well-being at work is imperative and holds substantial value in elucidating a more holistic understanding of teachers’ overall satisfaction and productivity in their professional roles.

SU has been hypothesized to have a positive relationship with well-being at work. Character strengths theory suggests that SU has a positive effect on well-being because people naturally engage in activities aligned with their inherent strengths [13]. A study on character strengths and subjective well-being confirmed that SU mediates the association between character strengths and hedonic well-being—in particular, that different character strengths could lead to different degrees of strength use [14]. In addition, SU also resonates with the Conservation of Resources (COR) theory, which recommends individuals to maintain, protect, and preserve their resources [15]. According to the COR theory, when people develop a resource surplus, they may experience positive well-being [16]. Moreover, the theory also posits that interventions that promote resource building and resource protection may improve well-being and prevent stress and burnout. Taken together, SU could be an important predictor of well-being at work. Consistently, SU has been found to be related to well-being in its general context [4], yet there remains a scarcity of research on its relationship to well-being at work, with only one study identifying a positive association between SU and work engagement—an indicator of eudaimonic well-being at work among college teachers [5]. Therefore, we aimed to further explore the association between strength use and both types of well-being within this specific occupational setting.

### 1.2. The Mediating Role of Basic Need Satisfaction

There are many studies that show people can increase their well-being by consciously engaging in a variety of positive activities. Lyubomirsky and Layous [17] proposed the positive-activity model to indicate when and why positive activity enhances well-being and examined the moderating and mediating mechanisms in the relationship. One of the important moderating factors is the person–activity fit, with SU happening to be a positive activity that brings the individual’s strengths into full utilization and fits a person to a high degree. Therefore, we believe that SU can improve individuals’ well-being by meeting basic psychological needs. However, limited research has been conducted on how this potential mechanism works through the lens of basic need satisfaction.

Among the theories in positive psychology, the most important theory on satisfaction of basic life needs is the self-determination theory (SDT). The SDT divides basic human needs into three components: autonomy (the need to authentically integrate experience and self-feeling), competence (the need to effectively control one’s environment) and relatedness (the need to foster a sense of connection and belongingness with others) [18,19,20]. It emphasizes basic psychological need satisfaction as a necessary condition for integrity, psychological growth, and well-being [21,22]. Several studies have demonstrated that the satisfaction of three basic psychological needs is related to overall well-being [23,24]. For example, Yang et al.’s research suggested that the satisfaction of basic psychological needs could account for the relationship between nature benefits and psychological well-being [25]. However, Deci proposes a model of eudaimonia based on the SDT. This model argues that behaving in ways that satisfy these three psychological needs reflects eudaimonic living, which thus contributes to high levels of EWB [26]. Therefore, we want to know if there is a difference in the impact of strength use on the two types of well-being under the mediating effect of basic need satisfaction. Moreover, within work and organizational settings, the satisfaction of these psychological needs has been positively related to employee well-being [27].

Furthermore, many studies have demonstrated the relationship between SU and basic need satisfaction [28,29,30,31,32]. For example, Kong and Ho investigate the role of strength use in the workplace by drawing on SDT to propose that strength use at work can yield performance benefits in terms of task performance and discretionary help [29]. SU can fulfill the need for competence by stimulating positive self-feedback and a sense of mastery [30]. Moreover, when employees use their strengths, they perceive their actions as spontaneous, which gives them a sense of autonomy [28]. Employees with a strong sense of autonomy possess the freedom to choose among work-related activities, thereby increasing the likelihood of engagement in many other areas in workplace [31]. In addition, individuals inherently seek respect and acknowledgement from others [31]. Using personal strengths allows individuals to perceive themselves as valuable contributors to a team while preserving their unique identity [32]. Together, these studies have suggested that SU may facilitate the satisfaction of basic needs.

In summary, SU, well-being, and basic need satisfaction seem to be correlated with each other. According to the positive-activity model, which suggests that positive activities such as SU may improve personal well-being by basic need satisfaction [17], it is reasonable to assume that basic psychological needs can mediate the association between SU and well-being at work. However, limited studies have explored this mediation effect. Therefore, our aim was to explore whether, within a work context, basic need satisfaction can act as a mediator in the relationship between SU and well-being.

### 1.3. This Study

This study explored the association between SU, basic need satisfaction, and well-being at work. Our research contributes to the existing literature in the following aspects. First, previous studies have focused on the use of strengths in employees in corporate settings but rarely on the impact of teachers’ use of strengths on well-being at work. Due to the scarcity of articles focusing on the impact of strength use on teachers’ well-being at work, we aimed to further explore the relationship between the two and their mediating mechanisms. Second, our study examined basic need satisfaction as a mediator between SU and well-being at work, and this has only been discussed by few researchers. Although the impact of basic need satisfaction on well-being has been proven by many studies, we investigated if basic need satisfaction can serve as a mediating variable to support the relationship between strength use and well-being. Finally, our study investigated both types of well-being, which was rare in the previous literature. We simultaneously studied the impact of strength use on two types of well-being and wanted to see if there was a significant difference in the impact of strength use mediated by basic need satisfaction on both types of well-being.

Building on prior literature, we formulated two hypotheses: (1) SU is positively correlated with teachers’ well-being at work, including both hedonic and eudaimonic well-being at work; (2) basic need satisfaction plays a mediating role between SU and well-being at work.

## 2. Methods

### 2.1. Participants and Procedure

This study was conducted in Shaanxi Province, with participants recruited online through WeChat or QQ. We randomly sampled participants from twelve departments within a university in Xi’an. Teachers willing to participate were directed to click on the provided link to complete an informed consent form and a formal questionnaire. Participants included teachers with varying years of service across different schools. In total, 384 teachers completed the questionnaire. After screening to eliminate respondents who did not provide careful responses (indicated by a filling time of less than 240 ms), a dataset of 374 valid responses was obtained, including 165 male teachers and 209 female teachers. A total of 83 participants reported working for less than five years, and 100 participants reported having more than 20 years of work experience.

### 2.2. Measures

#### 2.2.1. Strength Use

Govindji developed a questionnaire on the use of strengths in previous research [3], which was used to assess individual strength use with 14 items. The SU Scale was employed to assess the SU of teachers within their work environments. The adapted Chinese version of the SU scale demonstrated strong structural validity [33]. We employed a brief version of the SU Scale [3], which consists of 4 items rated on a 6-point scale ranging from 1 (never) to 6 (almost always), to measure teachers’ use of strengths. An example of the items was “I organize my job to suit my strong points”. A higher degree of strength use is reflected by a higher mean score of the entire scale. Cronbach’s alpha for this SU scale was 0.90.

#### 2.2.2. Basic Need Satisfaction

The most commonly used measure for basic need satisfaction is the Basic Psychological Needs at Work Satisfaction Scale [34]. We employed a brief version of the scale with 9 items, including autonomy satisfaction (AS), competence satisfaction (CS), and relatedness satisfaction (RS) (e.g., “I feel very competent when I am at work”; “In my job, I have many chances to show how capable I am”; “There is much opportunity for me to decide for myself how to go about my work”). Some items are reverse-scored. The scale adopted Likert’s five-point scoring method, ranging from 1 (highly disagreed) to 5 (highly agreed), and the scale demonstrated intracultural consistency. Cronbach’s alpha coefficients for the three subscales were 0.86, 0.72, and 0.77, respectively.

#### 2.2.3. Hedonic and Eudaimonic Well-Being at Work

To assess well-being at work, we employed two scales. The first was the Flourishing Scale. Diener proposed a scale that contained 8 items (e.g., I am competent and capable in the activities that are important to me) [35] to describe important aspects of human functioning, ranging from feelings of competence and holding positive relationships to having purpose and meaning in life. We introduced this scale into the work context. The scale uses Likert’s seven-point scoring method, which ranges from 1 (strongly disagreed) to 7 (strongly agreed), for expressing the teacher’s happiness at work. A higher level of eudaimonic well-being at work is reflected by a higher mean score of the entire scale. This scale’s Cronbach’s alpha was 0.91.

The next was hedonic well-being. We measured hedonic well-being by using the Positive and Negative Emotions Scale [36]. Words such as “relaxed”, “calm”, and “satisfied” measure positive affects, while words such as “anxious”, “frustrated”, and “nervous” measure negative affects [37]. A five-point Likert scale was used, with a score of 5 indicating “extremely” consistent with the respondent’s characteristics and a score of 1 indicating “very inconsistent or not at all consistent” with the respondent’s characteristics. Cronbach’s alpha coefficients for PA and NA were 0.875 and 0.907, respectively.

### 2.3. Data Analysis

To explore the association among strength use, well-being at work, and basic need satisfaction, we used SPSS 26.0 and Mplus 8 to conduct data analysis. First, Pearson’s correlations and descriptive statistics were calculated for the variables using SPSS 26. Then, a two-step approach was used to examine mediating effects. The initial evaluation of the measurement model aimed to gauge how well each latent variable was captured by its respective indicators. If the measurement model was accepted, we proceeded to test the structural model utilizing maximum likelihood estimation in Mplus 8. To address potential measurement errors associated with multiple items of latent variables, three-item parcels were formed separately for eudaimonic well-being at work.

### 2.4. Ethics Statement

Informed consent was obtained and signed for each participant at the beginning of this study.

## 3. Results

### 3.1. Correlations among SU, Well-Being at Work, and Basic Psychological Need Satisfaction

The means, standard deviations, and correlation coefficients among SU, well-being at work, and basic psychological need satisfaction are demonstrated in Table 1. Pearson’s correlation analysis was conducted to study the relationships among the variables. As expected, SU was significantly correlated with basic psychological need satisfaction. Meanwhile, SU was also positively correlated with both hedonic and eudaimonic well-being at work. In addition, basic psychological need satisfaction was positively associated with both hedonic and eudaimonic well-being at work as well.

### 3.2. Measurement Model

Four latent constructs were involved in the measurement model (SU, basic psychological need satisfaction, hedonic well-being at work, and eudaimonic well-being at work) together with 12 observed variables. We performed preliminary tests on the measurement model and obtained the following well-fitting data: *χ*^2^ (48, *N* = 374) = 83.27, *p* < 0.001; *CFI* = 0.988, *TLI* = 0.984; *RMSEA* = 0.044; *SRMR* = 0.026. The factor loadings of all the indicators on the latent variables were significant (*p* < 0.001), indicating that all the latent factors were well represented by their respective indicators. As shown in Table 1, all the latent factors from the measurement model were additionally significantly correlated (*p* < 0.001).

### 3.3. Structural Model

The direct path coefficient from the predictor (SU) to the criterion factors (hedonic well-being at work and eudaimonic well-being at work) were significant in the absence of mediating factors. The mediation model, mediated by satisfaction of basic psychological needs, showed a good fit to the data: *χ*^2^ (48, *N* = 374) = 83.27, *p* < 0.001; *CFI* = 0.988, *TLI* = 0.984; *RMSEA* = 0.044; *SRMR* = 0.026. The path coefficient from SU to hedonic well-being at work and eudaimonic well-being at work remained significant, suggesting that it was a partially mediated model.

We performed a bootstrap estimation with a sample of 1000 bootstrapped individuals to test the significance of the mediating effect of basic psychological need satisfaction. The fundamental concept behind the bootstrapping approach is that the estimates of indirect effects, which are the products of direct effects, typically do not adhere to normal distribution [38]. According to a few reports, the bootstrap approach produces the most precise confidence interval for indirect effects [39]. The results showed that the first pathway, indicating the indirect effect of SU on hedonic well-being through basic need satisfaction, was estimated at 0.348 (95% CI = [0.266,0.433]), and the indirect effect accounted for 63.85% of the total effect (0.545). Meanwhile, the second pathway, showing the indirect effect of SU on eudaimonic well-being through basic need satisfaction, was estimated at 0.449 (95% CI = [0.354,0.546]), and the indirect effect accounted for 71.73% of the total effect (0.629). These findings demonstrated significant indirect effects of SU on both hedonic well-being and eudaimonic well-being through basic need satisfaction. We also compared the standardized coefficients of the two mediation paths, and the results showed that the 95% confidence interval did not include zero (95%CI= [0.003,0.224]), which means that the impact of strength use on two types of well-being by basic need satisfaction was different. (see Figure 1).

## 4. Discussion

This study aimed to examine the relationship between teachers’ SU and well-being at work, along with the mediating effect of satisfying basic needs. We found that SU was significantly related to two types of well-being at work, mediated by basic need satisfaction.

Consistent with previous studies, SU was positively correlated with well-being at work [3,40,41,42]. This is also supported by SU theory [28,43], which claims that whenever individuals have the opportunity to leverage their strengths in the workplace, they become productive, energetic, and motivated [41]. These positive psychological states also contribute to employees’ proactive behavior [44], encouraging their active participation in the working environment [28] and consequently promoting overall well-being at work [41]. In addition, we found that a high level of SU can better satisfy basic psychological needs. The findings are consistent with previous studies, which have also found a positive correlation between SU and basic need satisfaction [45]. For example, Kong et al. applied SDT to determine the impact of workplace strength use on task performance and also demonstrated the relationship between strength use and basic need satisfaction [29]. This result also aligns with the content of the positive-activity model. The model suggests that positive activities can influence well-being through the mediating effect of ensuring basic psychological need satisfaction [17]. Individuals are likely to experience poorer well-being outcomes when positive activities produce negative (rather than positive) emotions, thoughts, and behaviors. Therefore, to increase the happiness of teachers in their work, it was essential to enhance their level of strengths utilization and meet their basic needs. Moreover, this study extends previous research on the effects of SU, showing that, other than overall work performance and well-being [46], using one’s strengths can also increase basic need satisfaction.

Notably, we studied the effects of SU on both hedonic and eudaimonic well-being at work in the same study. This is relatively rare in the prior literature, which has mostly focused on the effect of SU on hedonic well-being. Hedonic well-being considers happiness to be an experience of pleasure. Eudaimonic well-being, on the other hand, believes that happiness is not only pleasure but also the realization of human potential and the achievement and manifestation of human nature. In our study, we not only studied the impact of SU on teachers’ subjective emotions but also gained more insights into teachers’ psychological growth and development from strength use [7].

Furthermore, we found a mediating effect of the basic psychological need satisfaction factor between the use of strengths and teachers’ well-being at work. This is the first piece of evidence to demonstrate the importance of the satisfaction of basic psychological needs in the SU–well-being relationship in an organizational context. This result also corroborates the positive-activity model [17], which posits that positive activities such as SU may promote positive results, such as increased well-being at work, by satisfying basic psychological needs. This underscores the significance of not only focusing on SU but also addressing teachers’ basic psychological needs to enhance their well-being within the work environment. By ensuring the satisfaction of these basic needs, we can effectively stimulate teachers’ vitality and optimize their work efficiency. In addition, we tested the differences between the two mediating pathways and found that SU has a significantly greater impact on eudaimonic well-being than hedonic well-being at work by the mediating effect of basic need satisfaction. We believe that meeting basic needs can indeed affect two types of well-being, but it is more closely related to eudaimonic well-being, which is consistent with Deci’s theory [47]. The SDT may reveal this difference. SDT is a theoretical model that uses the eudaimonic definition as the central concept of well-being. The three basic needs of SDT are considered to be the three essential factors of well-being, but satisfying these three needs is, to a greater extent, consistent with eudaimonic well-being. The three basic needs must be met throughout the stages of life for people to experience a sustained sense of integration and well-being.

While our study contributes new insights, it is important to acknowledge several limitations. Firstly, it is crucial to recognize the inherent limitations of our study design. The questionnaire employed a cross-sectional study approach, which, by its nature, lacks the longitudinal data necessary to comprehensively trace changes in teachers’ well-being over time. This limitation restricts our capacity to establish more robust causal relationships. To overcome this, future research endeavors should prioritize the incorporation of long-term tracking data, facilitating a more nuanced understanding of the dynamic nature of teachers’ well-being and ensuring the acquisition of stable and reliable data and experimental results. Second, it is essential to consider the potential impact of our participant pool’s homogeneity, as our study exclusively included teachers from China. While this focus provided valuable insights into the well-being of Chinese teachers, it concurrently raised questions about the generalizability of our findings across diverse cultural contexts. To address this gap, future research initiatives should deliberately broaden their participant demographics to encompass teachers from varied cultural backgrounds. This approach not only enriches the scope and applicability of our conclusions but also contributes to a more comprehensive understanding of cross-cultural consistencies in the factors influencing teachers’ well-being.

For both teachers and their schools, our results have significant implications. First, teachers can use their strengths, which can increase teachers’ well-being at work and lessen negative emotions when in challenging situations at work. Second, to increase teachers’ well-being, managers should also foster an atmosphere that supports meeting their basic psychological needs. Intervention strength use based on SDT can be developed to improve the employees’ basic psychological need satisfaction and then increase well-being.

## 5. Conclusions

This study found positive correlations between SU and both hedonic and eudaimonic well-being at work, as well as the mediating effect of the satisfaction of basic needs on SU–well-being associations. Moreover, this study adds to the research of SU in the particular occupation of teaching and examines the distinction between hedonic and eudaimonic well-being. It deepens the understanding of the positive effects of SU, refines the understanding of its impact on well-being, fills theoretical gaps, and has real-life implications. These findings suggest that we can improve teachers’ well-being at work in two aspects. On one hand, enhancing the well-being of teachers can be achieved through the implementation of strengths-based intervention measures, tapping into their individual capabilities and strengths; on the other hand, an equally impactful approach involves shaping teachers’ overall well-being by addressing fundamental need satisfaction through targeted interventions, fostering an environment where their basic needs are met and supported.

## Figures and Tables

**Figure 1 behavsci-14-00095-f001:**
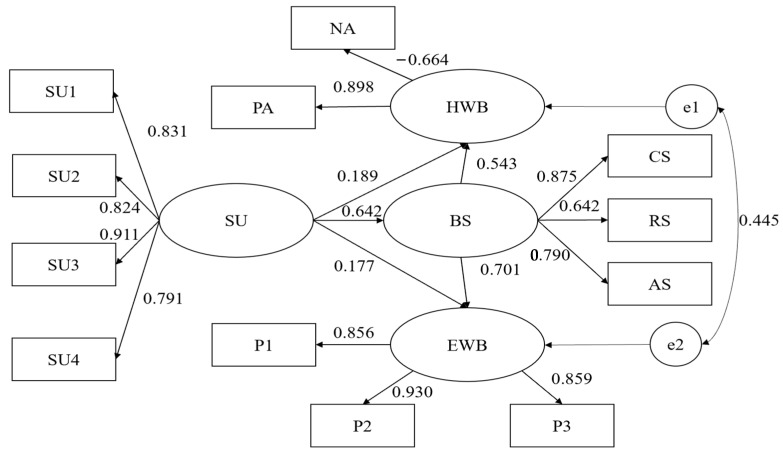
The finalized structure model (*N* = 374). Note: Factor loadings were standardized. SU1-SU4 = four items of strength use at work scale; CS, RS, and AS are the subscales of basic psychological need satisfaction; HWB = hedonic well-being at work; EWB = eudaimonic well-being at work; P1–P3 = three parcels of eudaimonic well-being; PA, NA = two parcels of hedonic well-being. All the path coefficients are significant (*p* < 0.001).

**Table 1 behavsci-14-00095-t001:** Descriptive statistics and correlations among the key variables.

Variable	*M* (*SD*)	1	2	3	4
1. Strength use	17.29 (3.53)	--			
2. Basic psychological need satisfaction	33.44 (4.85)	0.558 ***	--		
3. Hedonic well-being	4.91 (7.63)	0.491 ***	0.520 ***	--	
4. Eudaimonic well-being	44.06 (7.83)	0.582 ***	0.705 ***	0.618 ***	--

Note: *** *p* < 0.001.

## Data Availability

The data presented in this study are available on reasonable request from the corresponding author.

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
