# Peer review of "Strength Use and Well-Being at Work among Teachers: The Mediating Role of Basic Need Satisfaction"

_behavsci, 2024, doi:10.3390/bs14020095_

Round 1
Reviewer 1 Report
Comments and Suggestions for Authors
The study emphasizes the significance of teachers' well-being at work addressing the gap in research regarding the relationship between teachers' strengths utilization and their overall well-being at work. The manuscript highlights the importance of supporting educators in leveraging their strengths to enhance workplace satisfaction and mental health. Overall, the manuscript is well structured, the results are clearly presented, the discussion of findings is coherent and the conclusions are supported by the results. I have my own doubts that such a type of empirical analysis based on data gathered via questionnaire would have the potential to clearly provide reliable information on both eudaimonic and hedonic well being at the same time. Current literature highlights the difficulty of gathering consistent, objective, and reliable data via surveys on the topic of SWB since all these metrics only depend on the respondent’s perspective, which might be influenced even by the number or order of questions that the respondent needs to answer. Furthermore, attempting to encompass a wide array of data via questionnaires may compromise accuracy. This might explain the rarity of prior studies concurrently analyzing hedonic and eudaimonic well-being. I understand that a post-screening method based on filling time has been used to filter out inattentive respondents, but I wonder if this is enough. Undertaking this form of empirical analysis on SWB always poses inherent risks. Nonetheless, the potential benefits of concurrently analyzing the effects on both facets of well-being may justify taking this risk.
Another issue that I believe the authors need to address more thoroughly involves the study's statistical relevance which may comes into question due to the methodology employed in determining the sample and the method used to select participants. In my view, the selection process lacked randomness, thus the sample may not be a true representation of the broader population, leading to potential biases and skewed results. Moreover, the authors do not provide enough information on the sample which may raise concerns about the possibility of using the statistical methods employed by the study as well as about the generalizability of the findings to a more comprehensive scale. I understand that in this kind of study the main objective may be to capture a trend rather than to be statistically accurate, as the authors claim when they discuss the limitations of their study. However, I suggest that the authors provide more information about the sample process.
The authors should thoroughly check the manuscript to avoid typos and some citation inconsistencies. For example, at line 102 we have "deci believes" (without capital D) and without in text citation, s we don't know exactly which source is cited in this case. At line 110 we have the same text "Deci believes [30]" which makes reference to an incorrect source. Please check the authors for reference 30. Also, check reference 29.
Reviewer 2 Report
Comments and Suggestions for Authors
Abstract is well done and so with Literature Review. However some references are not very relevant but good enough. Please also double check your citations both internal and external ( references).
I would love to see clear research questions stipulated in this study.
Reviewer 3 Report
Comments and Suggestions for Authors
The paper explored the association between the three constructs of SU, basic needs satisfaction, and well-being of teachers at work. The study filled a gap by looking at both types of well-being at work, hedonic and eudaimonic. The quantitative methods were well done with significant results from my level of understanding.
I think the paper could be strengthened with more implications and examples of well-being actually in the workplace. In other words, point out the practical implications of the study a bit more.
I think this report is timely given teacher shortages and retention being a current issue. Do the references (particularly in text) need to be in APA style?).
Comments on the Quality of English Language
Line 26 -does this mean teachers? Confusing sentence.
Line 38 - remove “the”?
Line 40 - remove “as”?
Line 42 - remove “the”
Line 182 capitalization
Based on what I read about English language, I stopped listing these.
